# Histologic Analysis of Clinically Healthy Human Gingiva in Patients with Altered Passive Eruption

**DOI:** 10.3390/dj9030029

**Published:** 2021-03-06

**Authors:** Andrea Pilloni, Lorenzo Marini, Blerina Zeza, Amedeo Ferlosio, Rustam Aghazada

**Affiliations:** 1Section of Periodontics, Department of Oral and Maxillofacial Sciences, “Sapienza” University of Rome, 00161 Rome, Italy; andrea.pilloni@uniroma1.it (A.P.); rustam.aghazada@gmail.com (R.A.); 2Division of Periodontology, Department of Dentistry, Albanian University, 1001 Tirana, Albania; blerina.zeza@gmail.com; 3Anatomic Pathology, Department of Biomedicine and Prevention, Tor Vergata University of Rome, 00133 Rome, Italy; ferlosio@med.uniroma2.it

**Keywords:** anatomy, gingiva, health, histology, inflammation, periodontium, tooth eruption

## Abstract

The purpose of this study was to histologically examine the clinically healthy gingiva of patients with altered passive eruption (APE). Five patients with type 1 APE were enrolled. They underwent scaling and polishing and received oral hygiene instructions. After 6 months of supervised plaque control and uninterrupted gingival clinical health (Gingival Index (GI) = 0 and no Bleeding on Probing (BoP)), upper anterior teeth were surgically treated. During the surgical procedure, the excised gingival margin was collected to be histologically examined. In four out of five patients, signs of inflammation including spongiosis and neutrophil exocytosis could be found in the epithelium of the gingival sulcus. Ulceration with exposure of the lamina propria and inflammatory granulation tissue were evident in the most severe cases. Normal density and orientation of collagen fibers could be seen within the superficial and the deep portions of connective tissue, with an increase in size and number of the deep collagen fibers and a reduced laxity of the superficial ones. In conclusion, the clinically healthy gingiva of APE patients showed features compatible with persistent inflammation, possibly due to recurrent traumatisms caused by an incisally placed gingival margin.

## 1. Introduction

Altered passive eruption (APE) is a highly prevalent condition, occurring in 12.1% and 35.8% of the population depending on the diagnostic criteria [1,2]. It is defined as a situation in which “the gingival margin in the adult is located incisal to the cervical convexity of the crown and removed from the cemento–enamel junction of the tooth” [3]. APE was classified by Coslet et al. [4] into a type 1 or 2, depending on the location of the mucogingival junction in relation to the alveolar bone crest. While in the type 2 the distance between the gingival margin and the mucogingival line is within the normally accepted width (3.0–4.2 and 2.5–2.6 mm in the maxilla and in the mandible, respectively) and the mucogingival line is at the level of the cemento–enamel junction (CEJ), in the type 1 the band of attached gingiva is wider and the mucogingival line is apical to the alveolar ridge. Each type was further divided into a subgroup A or B, based on the location of the alveolar bone crest with respect to the CEJ. In the subgroup A the distance between the bone crest and the CEJ is normal (1–2 mm), whereas in the subgroup B the bone crest is at the level of or coronal to the CEJ.

Even though the gingiva of the patients with APE is usually healthy in the absence of plaque, several studies have related APE to periodontal health, assuming that APE is a possible risk factor for periodontal disease. Coslet et al. [4] theorized that APE type 2A (because the gingiva is unsupported by connective tissue fibers) and APE subtype B (because of the absence of collagen bundles of the gingival apparatus) are susceptible to periodontal breakdown. Prichard [5] hypothesized that the higher risk of developing periodontal disease in APE patients is due to an incisally placed gingival margin, which causes constant and repeated trauma and subsequent chronic inflammation of a bulbous marginal gingiva. Weinberg and Eskow [6] suggested that excessive keratinized mucosa may cause pseudopockets and interfere with adequate oral hygiene, determining the consequential plaque accumulation and inflammatory response. Volchanky and Cleaton-Jones [1] reported an association between APE and acute necrotizing ulcerative gingivitis. The authors argued that the association with this infectious disease could be due to the anaerobic environment predisposed by the presence of a deep gingival sulcus. Recently, Aghazada et al. [7] used a 21-day experimental gingivitis model to evaluate the inflammatory response of patients with APE compared to patients with normal gingival anatomy. In the presence of comparable amounts of plaque deposits, APE patients were observed to exhibit early development and delayed resolution of inflammation.

The gingival tissue structure is composed of a stratified epithelial tissue overlying a densely collagenous lamina propria. The epithelium facing the oral cavity constitutes the oral epithelium while the portion facing the tooth represents the sulcular epithelium, which continues with the junctional epithelium. The lamina propria comprises the supra-alveolar fiber apparatus, blood and lymphatic vessels and nerves. In the last decades, an attempt was made to distinguish between strictly normal and clinically healthy gingiva [8]. While strictly normal gingiva was considered an artifact that could be present under experimental circumstances, clinically healthy gingiva could exist in the presence of a physiological microbial and antigenic challenge. The clinically healthy gingiva shows a defensive architecture with the following characteristics: (i) a sulcular and a junctional epithelium with enlarged intercellular spaces, in which a considerable number of neutrophiles and crevicular fluid are present; (ii) a small area of macrophages, lymphocytes and plasma cells in the context of the lamina propria and a subepithelial plexus of venules [9]. On average, buccal gingiva is composed of 27% oral epithelium, 4% junctional epithelium and 69% connective tissue [10]. In the context of the 2017 World Workshop on the Classification of Periodontal and Peri-Implant Diseases and Conditions, a great effort was made to review the clinical and histological determinants of periodontal health. In fact, defining periodontal health was considered of fundamental importance to have a common reference point for evaluating periodontal disease and defining significant treatment outcomes [11]. Based on previous studies, it was stated that, in humans, a condition of pristine or clinically healthy gingiva with normal anatomy, even after a 6-month period of supervised oral hygiene practices, always histologically presents a slight inflammatory cell infiltrate [12,13]. It was reiterated that this is due to polymorphonuclear leukocyte physiological surveillance rather than a pathological process [11].

To the best of our knowledge, no study histologically described a clinically healthy gingiva in patients with APE. However, this information could be relevant since it could elucidate if in patients with this condition there are histological findings that could justify the above-mentioned assumptions of a higher susceptibility for periodontal disease. Therefore, the aim of this study was to histologically examine the gingival margin excised during the surgical treatment of patients with APE.

## 2. Materials and Methods

### 2.1. Study Design

For this histological human study, five patients requiring surgical treatment for altered passive eruption of the teeth comprised in the second sextant were enrolled. In the context of the surgical procedures, the gingival collar excised after the submarginal and sulcular incisions was collected to be histologically examined (Figure 1).

The present investigation was conducted at the Section of Periodontics of the Department of Oral and Maxillofacial Sciences, “Sapienza” University of Rome, Rome, Italy from June 2017 to December 2017.

### 2.2. Ethical Consideration

The study protocol was approved by the Ethical Committee of “Sapienza” University of Rome on 11 November 2015 (Prot. 2448/15). All patients signed a written informed consent document prior to participation. The investigation was performed in accordance with the Declaration of Helsinki of 1975, as revised in 2013 [14].

### 2.3. Participants

Patients with type 1 and subtype A/B altered passive eruption affecting the upper anterior teeth requiring surgical treatment for esthetic concerns were selected for this study, according the following exclusion criteria: (1) <18 years; (2) presence of systemic diseases; (3) cigarette consumption; (4) presence of any edentulism in the anterior upper arch; (5) presence of attachment loss; (6) presence of probing depth (PD) > 3 mm; (7) intake of drugs with a known effect on the gingival inflammatory response to oral biofilm (e.g., phenytoin, calcium channel blockers and cyclosporine); (8) history of periodontal surgical treatment; (9) pregnant or lactating; (10) the CEJ was not measurable with a periodontal probe; (11) presence of gingival overgrowth/hyperplasia, or inflammation; and (12) presence of anatomical alterations of the tooth crown (e.g., restorations, attrition, or traumatic injury).

Diagnosis of APE was performed clinically and confirmed radiographically. Preliminarily, patients with teeth with short clinical crowns and excessive gingival display (EGD) were examined. Then, a differential diagnosis to distinguish APE from other conditions that present comparable clinical features was accomplished (e.g., hypermobile upper lip, short upper lip, teeth attrition, excessive upper jaw growth and dentoalveolar upper extrusion). For the clinical investigation, a periodontal probe was employed. If the distance from the gingival margin to the CEJ was > 2 mm, the diagnosis of APE was made. [2,7]. For the radiographic analysis, a long-cone, parallel technique, periapical radiograph was used together with a gutta-percha cone to detect the gingival margin. The magnification of the image was adjusted by matching the real length of the gutta-percha cone with its length on the radiograph. The true length of the anatomical crown was estimated by measuring the distance between the CEJ and the gingival margin. The diagnosis of APE was confirmed if the length of the clinical crown (gingival margin to incisal edge) was significantly different (≥3 mm) to the length of the radiographic crown (CEJ to incisal edge) [15]. To discriminate between APE type 1 and 2, the attached gingiva was examined. Type 1 APE was diagnosed when the band of keratinized gingiva (from mucogingival junction to the gingival margin) was wider than a normally recognized mean width of 3.0–4.2 mm in the upper arch [16,17]. To distinguish between APE subtype A and B, “bone sounding”, under anesthesia, was used. A diagnosis of APE subtype A was made when the transgingival probing revealed a subgingivally positioned CEJ. Furthermore, the diagnosis of APE subtype A was certified when two distinct lines were identified in the radiographs (the most apical for the bone crest and the other for the CEJ) [15].

### 2.4. Pre-Surgical Procedures

Prior to the surgical treatment, patients underwent professional scaling and polishing to achieve gingival health and standardize baseline conditions. A medium toothbrush (Elmex Inter X, GABA International AG, Muenchenstein, Switzerland), standard toothpaste (Elmex, GABA International AG) and unwaxed dental floss (Elmex, GABA International AG) were provided. Polishing and oral hygiene instructions were repeated until patients were clinically healthy, as characterized by the absence of Bleeding on Probing (BoP) [18] measured on 6 sites and a Gingival Index (GI) [19] score of 0. Both of these assessments were made at the level of the upper anterior teeth (canine to canine). Patients were scheduled to have follow-up visits at 6 months and were carefully supervised for plaque control and optimal gingival health. After 6 months of uninterrupted clinical health, they were surgically treated.

### 2.5. Surgical Procedures

As suggested by Garber and Salama [20], in cases of altered passive eruption type 1A, a simple gingivectomy to expose the hidden anatomy of the crown was performed; in the cases of type 1B, an apically repositioned full-thickness flap with osseous resective surgery was made. In all of the cases, a first submarginal incision was made on the buccal surface of each papilla, leaving the interproximal tissue totally intact. Then, a sulcular incision was executed. The remaining gingival collar was excised with a periodontal curette. The removed tissue was immediately fixed in 10% neutral buffered formalin and the surgical treatment was continued and finally completed.

### 2.6. Histological Analysis

After being fixed in 10% neutral buffered formalin for 24 h, the samples were oriented through the identification of the oral epithelium and perpendicularly sectioned along the longitudinal diameter by 2 mm cuts. Each slide comprised the lining epithelium and the underlining lamina propria. All samples were embedded in paraffin wax and serial sections of 4 µm were cut at different levels and subsequently stained from each block with haematoxylin and eosin. The slices were examined under light microscopy with Nikon Eclipse E1200 and pictures taken by means of a Nikon camera system.

Microscopically, the following have been evaluated: (i) the integrity of epithelium or the presence of erosion/ulceration; (ii) the presence and characteristic of inflammatory infiltration (i.e., acute with neutrophils or chronic with lymphocytes and plasma cells); (iii) the orientation of collagen fibers and characteristic of capillary vessels have been considered.

## 3. Results

Five patients were treated, and the gingival collar excised during each surgical procedure was histologically examined. The study population consisted of four females and one male, aged 19 to 27 years with a mean age at baseline of 22.75 years ± 3.03. Two patients received a diagnosis of type 1 subgroup A APE, whereas three received a diagnosis of type 1 subgroup B.

At the 6-month follow-up visits and the day of the surgery, all patients exhibited GI = 0 and no BoP measured on six sites at the level of the upper anterior teeth.

Four out of five patients showed histological features comparable to gingivitis with different levels of severity.

### 3.1. Epithelial Tissue

The sulcular epithelium exhibited characteristic signs of inflammation comprising neutrophil exocytosis and spongiosis (Figure 2a,b).

Ulceration with exposure of the underlying lamina propria and proliferation of capillaries, distinctive signs of an inflammatory granulation tissue, were evident in the most severe cases (Figure 3).

The gingival epithelium exhibited areas of reactive hyperplasia with increased expression of the rete pegs and acanthosis, associated with varying degrees of intensity of a chronic lympho-plasmacellular inflammatory infiltrate (from perivascular to diffuse infiltrates) (Figure 4).

### 3.2. Connective Tissue

The collagen fibers maintained a normal orientation and a regular difference in density according with their position in the connective tissue. In particular, in the deep layer of the sub-epithelial connective tissue, collagen fibers were denser and more parallel to the gingival epithelium. The less numerous and thinner fibers of the superficial layer showed a “sunburst” pattern with respect to the root surface and the alveolar bone but with a perpendicular fashion related to the gingival lining epithelium (Figure 2a). In the superficial layer of the connective tissue there were elongated capillaries, while in the deep layer there were arterioles with orientation parallel to the collagen fibers. Particular aspects could be found in the increased number and size of the deep collagen fibers, due to a substantial sclerotization (Figure 5), and the lower laxity of the superficial ones. Moreover, in the sclerotic areas, a reduced number of vessels were present.

## 4. Discussion

Periodontal health could be thought of as a stable periodontium that functions comfortably in a person with psychological and social well-being about their mouth [21]. Biological, environmental, systemic, economic, social and psychological factors could all have an influence on an individual’s periodontal health [21].

APE was suggested to be an anatomical condition that could negatively affect periodontal health [1,4,5,6]. However, only one clinical study was carried out to clarify the hypothesis of a higher predisposition for periodontal diseases in patients with this condition. Aghazada et al. [7] conducted a study using an experimental gingivitis model and comparing patients with APE to patients without APE. The first group of patients was more prone to develop gingival inflammation when plaque was allowed to accumulate on their teeth than the second. In addition, once instructed to resume oral hygiene procedures, the resolution of the inflammation took longer in the APE group than in patients with normal gingival anatomy.

Our investigation is the first histological study performed to identify specific features of the clinically healthy gingival margin of the patients with APE, aiming to provide further information to understand this little-known clinical entity. In the literature, there are few studies on histologic analysis of human gingiva with normal anatomy. Brecx et al. [12] evaluated the cellular composition of developing infiltrated connective tissue in volunteers enrolled in a 21-day experimental gingivitis trial. As the clinical parameter for inflammation (GI) increased, the infiltrated connective tissue showed a substantial increase in lymphocytes (from 17.0% to 29.9%) associated with a reduction in the numerical density of fibroblasts (from 48.1% to 34.9%). No significant differences in the numerical density of polymorphonuclear leukocytes was observed (20.8% to 22.6%). In a second study, the gingival tissue of five dental hygienists with clinical indices for plaque and GI close to zero was analyzed at 0, 1, 4, and 6 months. During this long-standing optimal oral hygiene regime, the volumetric density of infiltrated connective tissue progressively decreased. In particular, the numerical density of lymphocytes within the infiltrate decreased considerably (from 18.4% to 5.6%), whereas the numerical density of fibroblasts increased (from 57.7% to 71.0%). As in the previous investigation, the numerical density of polymorphonuclear leukocytes was rather stable (from 20.6% to 17.7%) [13]. Both these studies indicated that a small inflammatory cell infiltrate is always present in clinically healthy gingiva with normal anatomy.

In the present study, the increase in the number and size of the deep collagen fibers, due to a considerable sclerotization, and the lower laxity of the superficial fibers did not seem to substantially differ from the studies by Brecx et al. [12,13], probably due to the prolonged status of clinical health. However, as an alternative explanation, there could be an increase in collagen synthesis or abnormal fibroblast function in patients with APE. On the other hand, in four out of five cases, the histological analysis of the epithelium of patients with APE showed features compatible with gingivitis which contrast with the condition of physiological immune surveillance observed in the above-mentioned investigations [12,13]. This exacerbated inflammatory response at the epithelial level could be justified, in the absence of plaque deposits and pseudopockets, by an incisally placed gingival margin which reduces the defensive capacity of the periodontium against recurrent traumatism, as proposed by several studies [22,23,24]. The results of this analysis could represent a possible explanation for the assumed predisposition of patients with APE for periodontal pathologies. Even in the absence of microbial biofilm-induced inflammation, the APE condition may create a “circle” that would maintain a gingival irritation over time with subsequent difficulties in maintaining domiciliary oral hygiene and further plaque accumulation.

In this study, the status of clinical health was ensured by the absence of BoP and GI = 0 from baseline until the surgical treatment of patients. Moreover, patients did not have probing depths > 3 mm or attachment loss. These criteria agree with the definitions of periodontal health proposed by the 2017 World Workshop on the Classification of Periodontal and Peri-Implant Diseases and Conditions [11]. A time lapse of 6 months occurred from the moment that patients were clinically healthy and the surgical treatment was selected, since it was observed that a persistent excellent oral hygiene condition is essential for any histological decrease in the inflammatory infiltrate [13].

Treatment of APE is basically surgical and consists of a crown-lengthening procedure. Two different approaches were described by Gaber and Salama [20] depending on the APE subtype and subgroup. In the type 1 subgroup A, the preferred option is the gingivectomy. In the type 2, the elective treatment is an apically repositioned full-thickness flap with, in the subgroup A, osseous resective surgery. Literature regarding the treatment of APE is scarce and merely discusses the outcomes of surgical procedures aimed at improving aesthetics. In this regard, Cairo et al. [25] showed that a carefully pre-operative planning and periodontal plastic surgery with or without osseous resective surgery can achieve predictable results. To date, no studies evaluated the potential beneficial effect of the treatment of APE on periodontal health. However, the findings of the present histological analysis are in agreement with the outcomes of our previous clinical study [7] and suggest that surgical restoration of normal gingival anatomy should be considered to reduce the increased tendency toward plaque-induced inflammation in patients with this condition.

The analysis was not able to address any specific cause for the histological findings and was not able to detect any differences among patients based on age, gender, and APE subgroups. In fact, all patients showed similar histological characteristics. This could be probably due to the main limitations of the present study: (i) the enrollment of only five subjects for this investigation, even though the same number of patients was selected in a comparable research [13]; (ii) patients with type 2 APE were excluded from this study, because the surgical treatment of these patients would not allow for submarginal incisions and the removal of a gingival collar for the histological examination of the gingival margin, lacking an adequate width of keratinized gingiva; (iii) the absence of a control site or group with gingiva with normal anatomy. For these reasons, case-control studies with a larger sample size are needed to confirm the results of the present study. Furthermore, the investigation of the biomolecular features of healthy, inflamed, wounded and unwounded human gingiva in APE patients is encouraged, to clarify if the healing process is somehow impaired in this specific population, leading, for example, to keloids or scar formation.

## 5. Conclusions

In their limits, the results of this first histologic study of the gingiva of APE patients show signs of inflammation even after an extended time interval of uninterrupted periodontal health.

## Figures and Tables

**Figure 1 dentistry-09-00029-f001:**
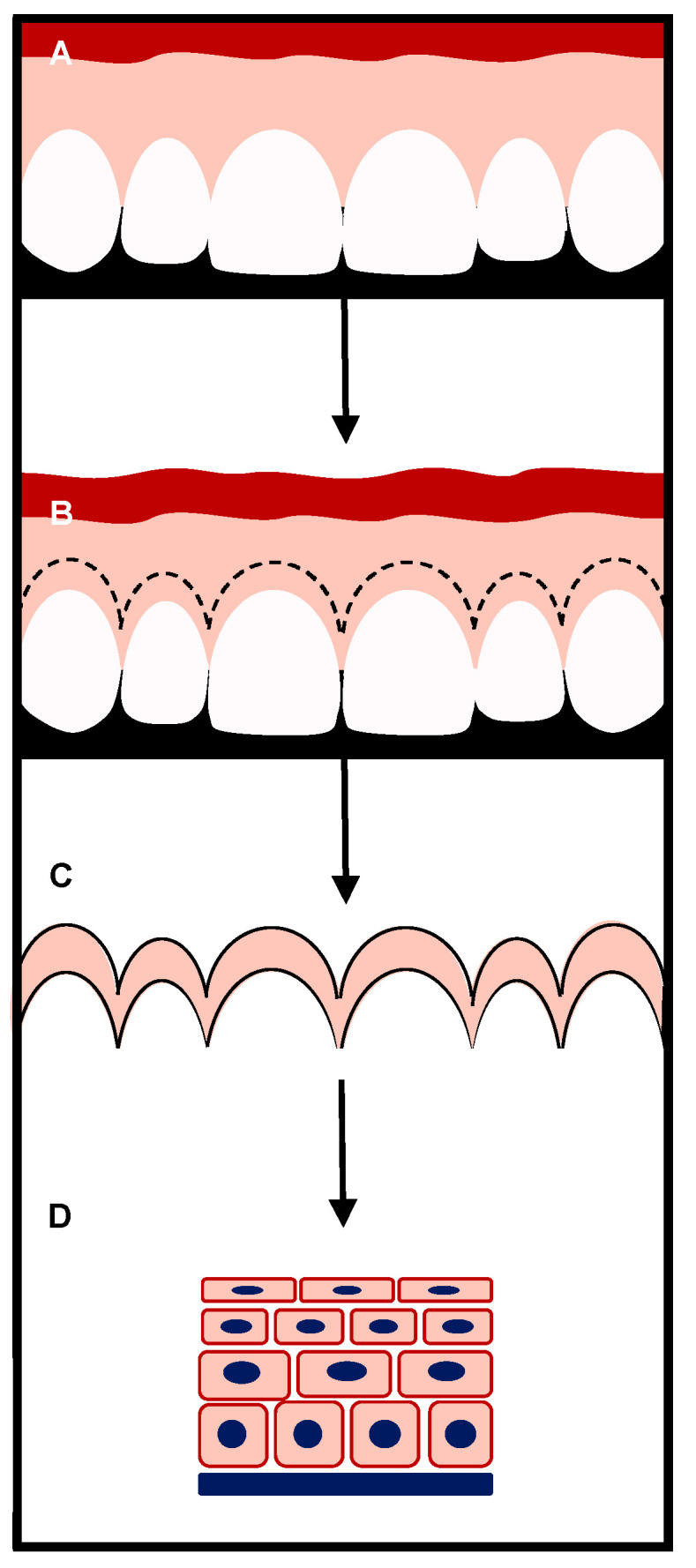
Study design. (**A**) Patients with type 1 altered passive eruption (APE) were selected. (**B**) After 6 months of supervised plaque control and uninterrupted gingival clinical health, patients were surgically treated. Submarginal and sulcular incisions were performed. (**C**) The gingival collar comprising the gingival margin was removed and fixed in 10% neutral buffered formalin. (**D**) The excised tissue was histologically analyzed.

**Figure 2 dentistry-09-00029-f002:**
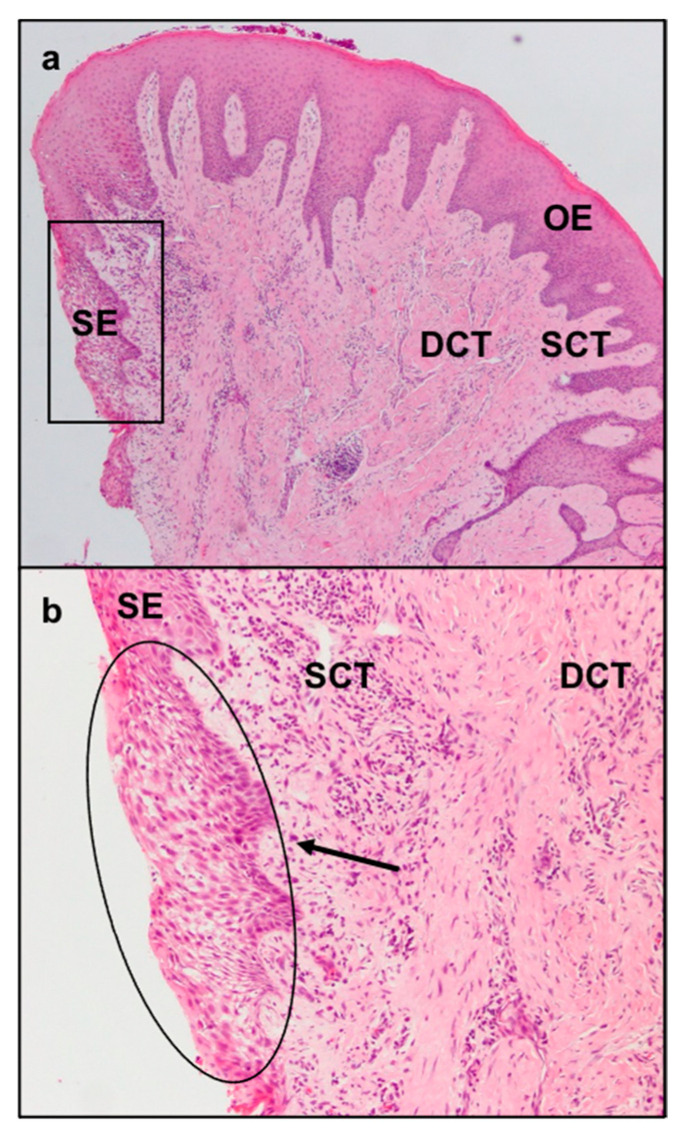
Light microscopy of 4 µm sections of the epithelial and connective tissues of a patient with APE and stained with haematoxylin and eosin. (**a**) Histological section corresponding to the gingival margin at a magnification of 4×. (**b**) Enlargement of the rectangular area in A at a magnification of 10×. In the sulcular epithelium (circled area), note the typical signs of inflammation, such as spongiosis and neutrophil exocytosis. OE = oral epithelium; SE = sulcular epithelium; SCT = superficial layer of connective tissue; DCT = deep layer of connective tissue.

**Figure 3 dentistry-09-00029-f003:**
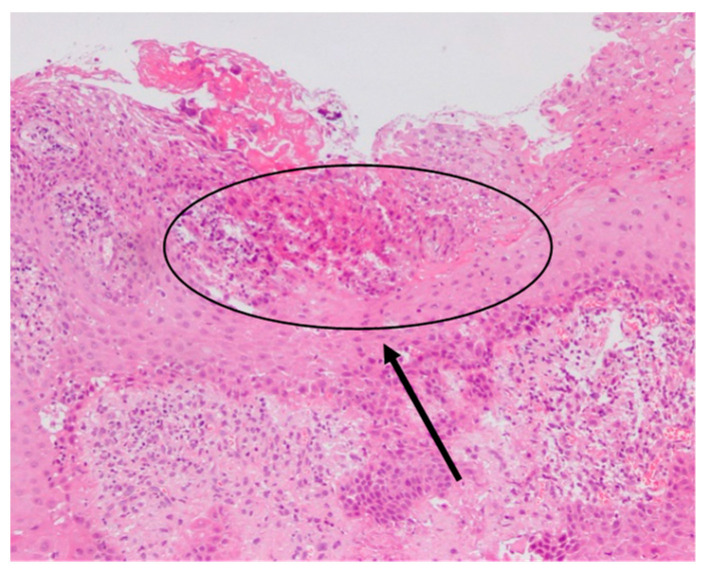
Light microscopy of 4 µm sections of the sulcular epithelial tissue of a patient with APE and stained with haematoxylin and eosin. In the circled area, it is possible to observe an extensive ulceration of the epithelium, a sign of severe inflammation at a magnification of 10×.

**Figure 4 dentistry-09-00029-f004:**
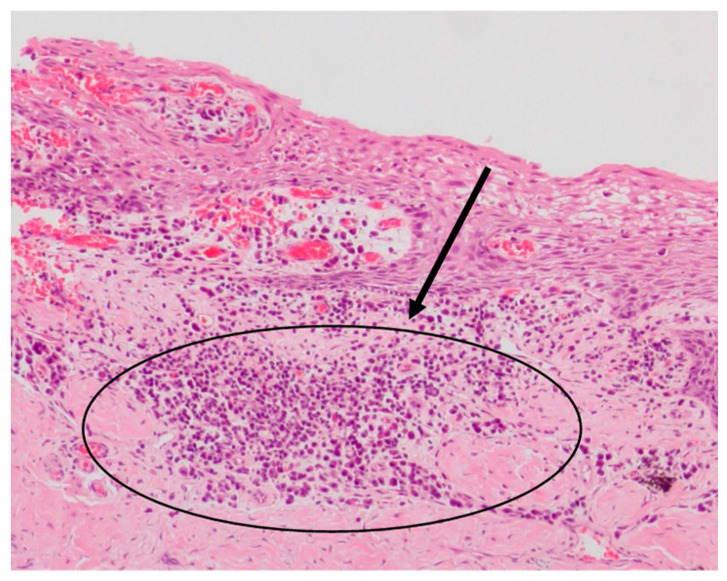
Light microscopy of 4 µm sections of the sulcular epithelial tissue of a patient with APE and stained with haematoxylin and eosin. In the circled area, an obvious chronic inflammatory plasma cell infiltrate is present, at a magnification of 10×.

**Figure 5 dentistry-09-00029-f005:**
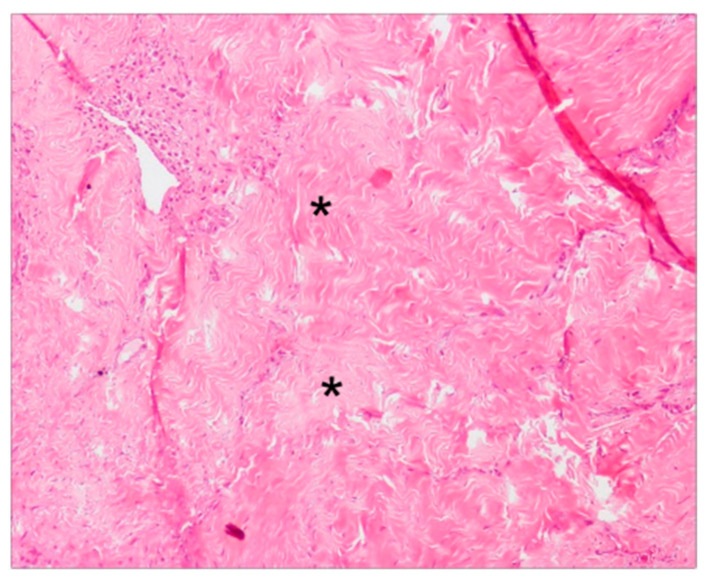
Light microscopy of 4 µm sections showing the substantial sclerotization of the deep collagen fibers, at a magnification of 10×. * = sclerotic areas.

## Data Availability

The data presented in this study are available on request from the corresponding author. The data are not publicly available due to privacy and ethical restrictions.

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
