# Peer review of "Histologic Analysis of Clinically Healthy Human Gingiva in Patients with Altered Passive Eruption"

_dentistry, 2021, doi:10.3390/dj9030029_

Round 1

Reviewer 1 Report

This manuscript is well-written and an interesting study to provide new knowledge in the field of dentistry. Although the topic is quite interesting, unfortunately, this study seems to be more suitable for case reports. There are limitations of sample size and group comparison. Authors are encouraged to consider submitting the revised manuscript as case reports.

1. [Page 1, Line 36] The authors mentioned that APE can be a risk factor for periodontal disease according to the type of APE. I recommend that the classification of APE should be provided more precisely to the readers as background information. So, please add the sentences or a paragraph.

2. [Figure 2] Please add caption in the figure 2B to improve readability. I think that the Figure 2B can be replaced with that with a higher magnification.

3. Please check and revise minor recommendations. Please refer to the notes in the pdf.

Reviewer 2 Report

" Histologic analysis of clinically healthy human gingiva in patients with altered passive eruption” It is very interest of histologically examine the clinical healthy gingiva of patients with altered passive eruption. However, there are corrections that are essential to meet the standard for publication. Please refer to the following comments.

  • I would like to know more about your thoughts on the clinical treatment of APE patients from the results of your study. Consideration of past treatment reports of APE patients is very useful to the author.

  • Please add about the limitations of your study. This study has a few cases.

Reviewer 3 Report

The authors present an ex vivo, focusing on histological characteristics of human gingiva in patients with altered passive eruption. This is an interesting topic and fits the journal's scope.

Since this study focuses on gingiva's histological aspects in patients with altered passive eruption, I suggest a brief description of gingiva's normal histology be included in the introduction section.

Lines 64-67: the authors stated that as to their knowledge, there are no histological studies on this topic. There are histological studies on patients with APE and periodontal diseases? If so, do the findings from those studies correlate with the obtained results?

Section 2.6: please refer to the orientation of the samples and what parameters were evaluated in the histological analysis

Results section: I find it of great interest if the authors could provide histological images of healthy gingiva from patients without APE. This would help the readers to better understand the study findings.

Figure 2: please add arrows or other symbols to the images to highlight the described features

Section 3.2: please provide images to illustrate the results of this section. Why did the authors not perform other colorations to better evidence the collagen fibers?

The discussion section needs to be more focused on discussing the differences between the obtained results and what is known about the healthy gingiva in patients without APE. Also, if these findings can or not support the theories described in the introduction section.

Lines 295-299: please move this information for the discussion section, since it is a hypothesizes and not a study conclusion

The manuscript references are mainly of old studies. There are no more recent studies on this topic?

Round 2

Reviewer 1 Report

My previous concerns were adequately corrected in the revised manuscript

I recommend to accept this manuscript for publication

Thanks for efforts. 

Author Response

Authors thank the reviewer for the previous comments and suggestions and for appreciating the current version of the manuscript. 

Reviewer 3 Report

The author's modifications improved the manuscript quality. Still, some changes can be made before the final manuscript version.  

Lines 187-189: the authors refer to the evaluation of “characteristic of capillary vessels”, but no reference to this appears in the results.

Please refer to figures 2a and 2b, instead of 2A and 2B, or change the figure letters.

Figure 5: use arrows or other symbols to evidence the sclerotization and the superficial fibers' lower laxity.
